# Intranasal Delivery of Gene-Edited Microglial Exosomes Improves Neurological Outcomes after Intracerebral Hemorrhage by Regulating Neuroinflammation

**DOI:** 10.3390/brainsci13040639

**Published:** 2023-04-08

**Authors:** Mengtian Guo, Xintong Ge, Conglin Wang, Zhenyu Yin, Zexi Jia, Tianpeng Hu, Meimei Li, Dong Wang, Zhaoli Han, Lu Wang, Xiangyang Xiong, Fanglian Chen, Ping Lei

**Affiliations:** 1Haihe Laboratory of Cell Ecosystem, Department of Geriatrics, Tianjin Medical University General Hospital, Tianjin 300052, China; 2Tianjin Geriatrics Institute, Tianjin Medical University General Hospital, Tianjin 300052, China; 3Tianjin Neurological Institute, Tianjin 300052, China

**Keywords:** intranasal delivery, miRNA-124, microglia, exosome, intracerebral hemorrhage, neuroinflammation, Gr-1^+^ myeloid cells

## Abstract

Neural inflammatory response is a crucial pathological change in intracerebral hemorrhage (ICH) which accelerates the formation of perihematomal edema and aggravates neural cell death. Although surgical and drug treatments for ICH have advanced rapidly in recent years, therapeutic strategies that target and control neuroinflammation are still limited. Exosomes are important carriers for information transfer among cells. They have also been regarded as a promising therapeutic tool in translational medicine, with low immunogenicity, high penetration through the blood-brain barrier, and ease of modification. In our previous research, we have found that exogenous administration of miRNA-124-overexpressed microglial exosomes (Exo-124) are effective in improving post-injury cognitive impairment. From this, we evaluated the potential therapeutic effects of miRNA-124-enriched microglial exosomes on the ICH mice in the present study. We found that the gene-edited exosomes could attenuate neuro-deficits and brain edema, improve blood–brain barrier integrity, and reduce neural cell death. Moreover, the protective effect of Exo-124 was abolished in mice depleted of Gr-1^+^ myeloid cells. It suggested that the exosomes exerted their functions by limiting the infiltration of leukocyte into the brain, thus controlling neuroinflammation following the onset of ICH. In conclusion, our findings provided a promising therapeutic strategy for improving neuroinflammation in ICH. It also opens a new avenue for intranasal delivery of exosome therapy using miRNA-edited microglial exosomes.

## 1. Introduction

Intracerebral hemorrhage (ICH) is a common disease that constitutes 15–20% of stroke, but the mortality rate is up to 50% [1]. Although the therapeutic strategies have advanced rapidly, the effective strategies to control the progression of the disease are still limited [2]. Generally, the pathological development of ICH is rapid and highly complex [3]. The mechanical disruption caused by parenchymal bleeding of the brain triggers a cascade of pathological changes, involving glial cell activation in regions adjacent to the hematoma and disruption of the blood-brain barrier (BBB) [4]. The infiltration of peripheral leucocytes into affected cerebral tissues through a dysfunctional blood-brain barrier leads to the mass production of pro-inflammatory cytokines [5]. The inflammatory cascade accelerates the formation of perihematomal edema (PHE) and amplifies cell death [6]. Hence, alleviating the inflammatory response emerged as a potential therapeutic target for acute intracerebral hemorrhage treatment.

Exosomes, a subtype of extracellular microvesicles (30–150 nm), mediate cell–cell communication through transporting biologically active cargo (including proteins, lipids, and nucleic acids) to recipient cells [7]. Increasing evidence has demonstrated that exosomes are involved in the maintenance of central nervous system homeostasis and that brain cell dysfunction can modulate the immune response through the release of exosomes [8]. Recent studies have revealed that the inhibition of endogenous exosome release augments neuroinflammation and brain injury following ICH. The versatility of exosomes in translational medicine, including applications in disease prevention, diagnosis, and treatment, has attracted wide attention [9,10,11,12]. Exosome therapy is superior to traditional treatment methods, having the characteristics of low risk of vascular obstruction, high penetration through the blood-brain barrier, and excellent biodistribution and biocompatibility [13,14]. Emerging studies focus on the role of exosomal miRNAs in the pathological processes of various diseases and attempt to elucidate the underlying mechanisms of treatment [15,16,17,18,19]. In our previous work, we have confirmed that upregulated miRNA-124 microglial exosomes inhibit neuroinflammation and neurodegeneration after traumatic brain injury [20]. In addition, miRNA-124 promotes microglia anti-inflammatory M2 polarization in vitro and alleviates neurodegeneration after repetitive mild traumatic brain injury via transferring exosomes into hippocampal neurons [21]. Of note, evidence suggests that exosomal miRNAs are crucial regulators in ICH [22]. Recent research indicates that exosomal let-7i from patients with multiple sclerosis regulates the pathological development through blocking the IGF1R/TGFBR1 signal pathways [23]. However, the potential impact of Exo-124 on brain injury and inflammation in acute intracerebral hemorrhage is still unclear. Considering the crucial role of enhancing exosomal miRNA bioactivity on neurological impairment, the present study focuses on investigating whether intranasal delivery of Exo-124 improves the neurological outcomes of bacterial-collagenase-induced ICH mice and exploring the underlying mechanism.

## 2. Materials and Methods

### 2.1. Animals

All experimental procedures were approved by the Tianjin Medical University Animal Care and Use Committee on 15 November 2021 (Protocol number: IRB2021-KY-272). Adult male C57BL/6J mice (8–10 weeks old) were purchased from the Chinese Academy of Military Science (Beijing, China). Animals were housed in animal care facilities under a standardized light–dark cycle with sufficient food and water.

### 2.2. Induction of ICH in Mice

The ICH mouse model was induced by bacterial collagenase as previously described [24,25]. Briefly, a stereotactic frame was designed to fix the mice after anesthesia with sodium pentobarbital (50 mg/kg). A 26-gauge needle was slowly inserted into the right striatum (2.3 mm lateral to the midline, 0.5 mm anterior to bregma, 3.6 mm depth below the surface of the skull). After that, 0.05 U TESCA buffer-diluted (Solarbio, Beijing, China) bacterial collagenase (type IV-S; Sigma-Aldrich, St. Louis, MO, USA) was infused at a rate of 0.5 μL/min through a micro-injection pump (KD Scientific, Holliston, MA, USA). Sham controls underwent the same procedures and were injected without drugs. After surgery, animals were housed in cages under observation with sufficient food and water.

### 2.3. Cell Culture and Transfection

BV2 microglia were purchased from BNCC (BeNa Culture Collection, BNCC337749, Henan, China) and cultured in DMEM/F12 complete culture medium with 10% Exo-depleted fetal bovine serum (FBS) (GIBCO Laboratory, Grand Island, NY, USA) at 37 °C. For the preparation of exosome-depleted fetal bovine serum (FBS), regular FBS was conducted by ultracentrifugation at 100,000 *g* for 18 h. The purity of cultured BV2 microglia was identified by immunofluorescence staining of Iba-1 (1:500,Abcam, Cambridge, Cambridgeshire, UK). To establish stable cell lines, lentivirus vectors-GFP (GeneChem, Shanghai, China) containing mmu-miR-124-1 and negative control miRNA (miRNA-NC) were transduced into BV2 microglia with transduction reagents (GeneChem) for 12 h according to the manufacturer’s instructions. A quantity of 2 mg/mL puromycin was used to screen the infected cells. The transfection efficiency of BV2 microglia cell lines was confirmed by qRT-PCR. 

### 2.4. Exosome Preparation and Identification

To enrich exosomes, we performed the method of ultracentrifugation as described previously [26]. Firstly, collected cell culture supernatants were prepared by centrifugations at 300× *g* for 10 min to remove free cells. For the removal of cell debris, the supernatants were rotated at 2000× *g* for 10 min at 4 °C. To remove cell pellets, the supernatants were spun at 10,000 *g* for 30 min at 4 °C. Subsequently, the supernatants were filtered with a 0.22 μm filter (Millipore-Sigma, St. Louis, MO, USA) to remove dead cells and larger particles. The exosomes were enriched by ultracentrifugation at 100,000× *g* for 70 min at 4 °C using a swing rotor (SW32Ti, XPN-100, Beckman, CA, USA). The freshly isolated particles were used for the identification of exosomes. The protein content of the exosome was quantified by bicinchoninic acid (BCA) assay (Solarbio, Beijing, China) following the manufacturer’s instructions. Briefly, exosome sample was added to 96-well plate incubating with the prepared BCA working reagent for 30 min at 37 °C and cooling to room temperature; the resultant purple color was measured at 562 nm. The morphology of the isolated particles was identified by transmission electron microscope (TEM, HT8700; Hitachi, Tokyo, Japan). The representative biomarkers of exosomes including CD63, CD81 (1:1000, Abcam, UK) and CD9 (1:1000, Abcam, UK) were detected by Western blot analysis. The size distribution of the particles in the precipitation was detected by Nano Particle Tracking (NTA).

### 2.5. Drug Administration

Mice were intranasally administered once a day with approximately 2.11 × 10^9^ ± 1.04 × 10^7^ particles/mL exosome for 3 consecutive days after ICH induction [21,27]. The freshly isolated exosomes were administered at the ICH surgery day. The follow-up experiments were performed using exosomes stored at 4 °C for short-term preservation. The storage temperature of exosomes for long-term preservation was −80 °C. Minimized freeze-thaw cycles are beneficial to maintain the characteristics and functions of exosomes [28]. For depletion of Gr-1^+^ myeloid cells in vivo, mice were intraperitoneally injected with anti-mouse Gr-1 monoclonal antibody (MAbRB6-8C5; BioXcell, Lebanon, NH, USA) with a dose of 250 μg at 1 day before and 1 day after ICH surgery. In vivo rat IgG2b (LTF-2; BioXcell, Lebanon, NH, USA) was used as isotype control [29,30]. For the verification of Gr-1^+^ cell depletion, flow cytometry was performed.

### 2.6. Neurological Behavior Assessment

As previously described, two investigators blinded to experimental groups performed neurological tests at days 1 and 3 after ICH induction [31]. A modified neurological severity score (mNSS) was conducted to evaluate neurological functions including motor functions (abnormal movement and muscle status), sensory functions (visual, tactile, and proprioceptive), reflex reactions, and balance functions. Briefly, the mNSS rates neurological deficits on a scale of 18, in which 0 represents normal, and a higher score indicates a more severe injury [32,33]; the mouse was given a corresponding score for abnormal behavior. To assess postural asymmetries, we performed a corner-turning test as previously described [24]. The mice were placed in a dark corner with an angle of 30° and were allowed to randomly turn around to exit the corner. The percentage of right turns was recorded in 10 repeated trials with an interval of more than 30 s [31]. For the foot-fault test, we designed a metal grid device of size 20 cm × 20 cm × 50 cm (length × width × height) with a 12 mm mesh. The mice were allowed to traverse the grid freely for 5 min. Foot fault was identified as forelimbs slipped into the grid holes. The percentage of foot faults to total steps was calculated.

### 2.7. Hematoma Volume Analysis

The measurement of hematoma volumes was performed as previously described [33]. In brief, brain coronal sections (8 µm thick) were cut from bregma −4 mm to bregma +2 mm with an interval of 1 mm. Six coronal slices of each mouse were stained with hematoxylin and eosin (H&E) for hematoma volume calculation. The hematoma areas were summed and multiplied by the section thickness. Image J (U.S. National Institutes of Health) software was used to analyze hematoma volume.

### 2.8. Brain Water Content Assessment

The brain water content assessment was carried out on day 3 after ICH onset as previously described [25]. In brief, the brains were carefully divided into three parts, including the bilateral cerebral hemispheres and the cerebellum. The brain tissues were dried for 24 h at 100 °C in an oven. The wet weight and dry weight of brain tissues were recorded before and after drying using an electronic balance. The brain water content was calculated by the following formula: (wet weight − dry weight)/wet weight × 100%. 

### 2.9. BBB Permeability

The measurement of BBB permeability was conducted on day 3 after ICH onset as previously described [31]. Briefly, the mice were intravenously injected with 2% Evans Blue dye (2 mL/kg, Sigma, St. Louis, MO, USA) 2 h before sacrifice. Blood perfusion is required to avoid the contamination of Evans Blue in blood. The ipsilateral hemisphere was collected and weighed to obtain the wet weight. After that, the brain tissues were homogenized with 1 mL formamide (Sigma, St. Louis, MO, USA) followed by incubation in a 70 °C water bath for 24 h. The supernatants were added into 96-well plates after centrifugation at 8000 rpm for 5 min. The optical density (OD) values of the supernatants and the standards were measured by a microplate reader (Thermo Fisher) at a wavelength of 630 nm. The following formula was used for calculation: EB content (µg/g wet brain) = EB concentration × formamide (mL)/wet weight (g). 

### 2.10. Immunofluorescence Staining 

For analysis of cell death after ICH, we performed TUNEL assays to determine whether treatment with Exo-124 reduces apoptosis in the peri-hemorrhage area of the brain section. In brief, frozen sections were washed with PBS and subjected to staining with TUNEL assay (Roche, Basel, Switzerland) for 2 h at 37 °C. After that, the sections were incubated with 4′,6-diamidino-2-phenylindole (DAPI, Abcam, Cambridge, Cambridgeshire, UK) for 7 min at room temperature to assess the nuclear morphology. The brain sections were captured by a fluorescence microscope (Olympus IX81, Tokyo, Japan). For the quantification of cell death, positive cells around the hematoma area from four random fields (six sections per mouse) were calculated. The data were presented as the positive ratio of TUNEL to DAPI [34]. 

### 2.11. Real-Time Polymerase Chain Reaction (RT-PCR)

According to the manufacturer’s suggested protocol, the total RNA of exosomes, cells, and brain tissues was extracted with an EasyPure miRNA Kit (ER601, Transgenic, Beijing, China). The concentration and purity of samples were quantified using a Nanodrop ND-2000 (Thermo Fisher Scientific, Waltham, MA, USA). Reverse transcription was carried out using a cDNA reverse transcription kit (Tian Gen, Beijing, China). The cycle threshold was detected by a Real-Time PCR Detection System (Bio-Rad, Hercules, CA, USA). All mRNA levels were normalized to glyceraldehyde 3-phosphate dehydrogenase (GAPDH). The primers of inflammatory mediators were designed and synthesized by Sangon Biotech (Shanghai, China). For miR-124 detection, cDNA reverse transcription and RT-PCR analysis were carried out using the Hairpin-it miR-124 RT-PCR Quantitation Kit (GenePharma, Shanghai, China). U6 was regarded as the control. The details of primer sequences are listed in Table 1. The data were analyzed with the 2^−△△ Ct^ method.

### 2.12. Western Blot Analysis

The extraction of total protein from ipsilateral brain tissues followed the established procedures as previously described [21]. The protein concentration was quantified by the BCA Protein Assay Kit (Thermo Fisher Scientific, Waltham, MA, USA) according to the manufacturer’s protocol. Briefly, normalized protein samples were loaded onto SDS-PAGE gels and transferred onto PVDF membranes (Millipore, Waltham, MA, USA). After that, the membranes were incubated with 5% skim milk to block nonspecific staining at room temperature for 2 h. Finally, the membranes were washed with TBST three times and incubated with primary antibodies overnight at 4 °C. GAPDH was used as the internal control. The primary antibodies were listed as follows: ZO-1 (1:1000; proteintech, San Diego, CA, USA), Occludin (1:1000; Abcam, Cambridge, Cambridgeshire, UK), Claudin-5 (1:1000; Abcam, Cambridge, Cambridgeshire, UK), CD9 (1:1000; Abcam, Cambridge, Cambridgeshire, UK), CD63 (1:1000; Abcam, Cambridge, Cambridgeshire, UK), CD81 (1:1000; Abcam, Cambridge, Cambridgeshire, UK), MMP9 (1:1000; Abcam, Cambridge, Cambridgeshire, UK) and GAPDH (1:1000; Cell Signaling Technology, Dallas, TX, USA). The next day, membranes were incubated with corresponding secondary antibodies (ZSGB-BIO, Beijing, China) at room temperature for 1 h. Finally, membranes were scanned with an imaging system (Bio-Rad, Hercules, CA, USA) and quantified using Image J 7.0 software (National Institutes of Health, Bethesda, MD, USA). 

### 2.13. Flow Cytometry

To analyze the counts of leucocyte infiltration and microglia in brains, we performed flow cytometry analysis as previously described [35]. Briefly, the brain tissues were cut into small pieces and digested in 1 mg/mL collagenase (Solarbio, Beijing, China) in PBS at 37 °C for 30 min, and then isolated cell pellets were collected after centrifugation at 2000 rpm for 5 min. After that, myelin was removed using 30% Percoll solution (GE Healthcare Bio-Science AB, Uppsala, Sweden) after centrifugation at 700 *g* for 10 min. The precipitation was collected and resuspended with 1% bovine serum albumin (BSA). The cell suspensions were stained with antibodies for 30 min. Blood samples were incubated with blood cell lysis (Solarbio, Beijing, China) for 20 min at room temperature. Cell pellets were collected after centrifugation at 2000 rpm for 5 min and stained with antibodies for 30 min [25]. All antibodies were obtained from BioLegend (San Diego, CA, USA) and listed as follows: CD45 (30-F11), CD11b (M1/70), CD3 (145-2C11), CD4 (GK1.4), CD8 (53–6.7), anti-NK1.1 (PK136), anti-CD19 (1D3), Ly6C (HK1.4), Ly6G (1A8) and Gr-1 (RB6-8C5). Flow Jo v10.4 software (Informer Technologies, Walnut Creek, CA, USA) was used to analyze the flow cytometry data.

### 2.14. Statistical Analysis

All data were analyzed using GraphPad Prism 9.0 software (GraphPad, La Jolla, CA, USA) and presented as the mean ± SD. One-way ANOVA followed by the Tukey post hoc test was used to compare data among three or more groups. *p* values less than 0.05 were considered significant.

## 3. Results

### 3.1. Characteristics of miRNA-124-Enriched Microglia-Derived Exosomes

For the identification of BV2 microglia, light microscopy was used to observe the morphology of the cultured cells. In addition, the cultured BV2 microglia was identified by immunofluorescence staining of Iba-1 (Figure 1A,B). To establish stable BV2 cell lines with upregulated miRNA-124 expression, BV2 microglia were transfected with lentivirus expressing miR-124-GFP or miRNA-NC and evaluated by fluorescence microscopy (Figure 1C). The TEM image indicated that the particles were round, with a diameter ranging from 30 to 200 nm (Figure 1D). The NTA results indicated that the peak diameter of the particles was 148 nm (Figure 1E). Western blot analysis showed that surface exosome markers, including CD9, CD63 and CD81, were highly expressed in the precipitate but expressed at low levels in the supernatant (Figure 1F). The transfection efficiency of miRNA-124 in microglia and microglia-derived exosomes was evaluated by RT–PCR (Figure 1G). We found that the expression of miRNA-124 was significantly elevated in microglia and its exosomes. The results indicate that the characteristics of the isolated extracellular vesicles were consistent with those of exosomes.

### 3.2. miRNA-124-Enriched Microglia-Derived Exosomes Attenuate Neurological Deficits, Brain Edema after ICH

Neurological deficits, the hematoma volume and the extent of brain edema were assessed to explore the effects of miRNA-124-enriched microglia-derived exosomes (Exo-124) after ICH onset. Reportedly, nasal administration has the advantages of noninvasiveness, brain targeting, and low risks of overintravenation or intracerebroventricular injection [36]. The mice were intranasally administered Exo-124, Exo-NC or vehicle once a day for three consecutive days after ICH induction (Figure 2A,B). Neurological function was evaluated on days 1 and 3 after the onset of ICH by the Neurological Severity Score (mNSS), foot-fault test and corner turning test. The hematoma volume and extent of brain edema were quantified on day 3 after ICH onset. We found that the neurological deficits were aggravated on day 1 and 3 after ICH. Compared with the PBS-treated group, the Exo-124-treated group showed a significant reduction in neurological deficits on day 1 and 3 after ICH (Figure 3A). In addition, we found that the hematoma volume was significantly augmented after ICH, and that there was no significant change in the Exo-124 group compared with the PBS-treated group (Figure 3B,C). Furthermore, ICH induction increased the brain water content, and Exo-124 treatment reversed the change after ICH (Figure 3D). These findings suggest that Exo-124 is beneficial in alleviating neurological deficits and reducing brain edema after ICH induction.

### 3.3. miRNA-124-Enriched Microglia-Derived Exosomes Attenuate BBB Damage after ICH

Evidence indicates that BBB breakdown is a crucial indicator of secondary brain injury after ICH onset [37]. To determine whether Exo-124 is involved in protecting BBB integrity after ICH induction, we measured the extravasation of Evans Blue and the expression of tight junction proteins. The results showed that ICH induction significantly increases the leakage of Evans blue dye and induces a decrease expression of tight junction proteins compared with the sham group. Exo-124 significantly ameliorated Evans blue dye leakage (Figure 4A,B). The levels of tight junction proteins, including ZO-1, claudin-5 and occludin, were significantly increased in the Exo-124 group compared with the PBS-treated group (Figure 4C,D). Therefore, we conclude that Exo-124 attenuates BBB damage after ICH.

### 3.4. miRNA-124-Enriched Microglia-Derived Exosomes Reduce Cell Death and Inhibit Brain Inflammation after ICH

To assess the impacts of Exo-124 on neuroinflammation, the ipsilateral hemisphere was collected from mice with collagenase-induced ICH, and the expression of inflammatory mediators was measured using RT–PCR. The production of TNF-α, IL-1β, and IL-6 was significantly increased in ICH mice compared with sham controls. However, the gene expression of IL-10 showed the opposite trend. Our data suggest that Exo-124 treatment reduces the expression levels of IL-1β, IL-6, and TNF-α but inhibits the increase in the expression of IL-10 (Figure 5A–D). Previous studies have reported that MMP9 levels are elevated in perihematomal tissue from ICH patients and positively associated with exacerbation of neurological dysfunction [2]. We found that the expression of MMP9 was significantly elevated in mice after ICH induction compared with the sham controls. The expression of MMP9 was decreased in Exo-124-treated group compared with the PBS-treated group (Figure 5E,F). In addition, immunostaining was performed to evaluate cell death in the experimental groups. The number of TUNEL-positive cells in the perihematomal area was significantly decreased in the Exo-124-treated group compared with the PBS-treated group (Figure 5G,H). These results demonstrate that Exo-124 treatment exerts a protective effect by inhibiting neuroinflammation and reducing cell death after ICH.

### 3.5. miRNA-124-Enriched Microglia-Derived Exosomes Reduce Immune Cell Infiltration into the Brain after ICH

To further explore the potential mechanisms by which Exo-124 confers neuroprotection after ICH onset, we examined leukocyte infiltration in the brain by flow cytometry. We found that the counts of leukocytes including neutrophils (CD45^high^CD11b^+^Ly6G^+^), monocyte/macrophages (CD45^high^CD11b^+^Ly6C^+^), CD4^+^ T cells (CD45^high^CD3^+^CD4^+^) and natural killer cells (CD45^high^CD3^−^NK1.1^+^) were increased in the brain after ICH induction. The result was consistent with the previous findings [38,39]. The counts of neutrophils (CD45^high^CD11b^+^Ly6G^+^) and monocyte/macrophages (CD45^high^CD11b^+^Ly6C^+^) were significantly decreased in the mice receiving Exo-124. However, there was no significant difference in the number of microglia (CD45^int^CD11b^+^), T cells (CD45^high^CD3^+^CD4^+^/CD45^high^CD3^+^CD8^+^), natural killer cells (CD45^high^CD3^−^NK1.1^+^) or B cells (CD45^high^CD3^−^CD19^+^) between the PBS-treated group and the Exo-124-treated group (Figure 6A–D). Our findings indicate that Exo-124 reduces leukocyte infiltration into the brain after ICH onset.

### 3.6. Gr-1^+^ Myeloid Cells Are Involved in the Protective Effect of Exo-124

Previous studies have reported that Gr-1^+^ myeloid cells play a role in BBB disruption and brain inflammation after ICH [40]. As the numbers of neutrophils and monocytes were significantly decreased in ICH mice after Exo-124 administration (Figure 6C), we postulate that Exo-124 may confer protection against ICH by inhibiting Gr-1^+^ cell infiltration. To verify the role of Gr-1^+^ myeloid cells in the protective effect of Exo-124, we depleted Gr-1^+^ myeloid cells using an anti–Gr-1^+^ mAb. (Figure 7A). First, we confirmed the depleted efficiency of Gr-1^+^ cells from the blood after anti–Gr-1^+^ mAb injection using flow cytometry. We found that the percentage of Gr-1^+^ cells was significantly decreased in mice injected with the anti–Gr-1^+^ mAb (Figure 7B,C). Gr-1^+^ myeloid cell depletion diminished the neuroprotective effect of Exo-124 treatment after ICH injury (Figure 7D), suggesting that Gr-1^+^ myeloid cells contribute to the protective effect of Exo-124.

## 4. Discussion

The present study demonstrates that Exo-124 significantly attenuates hemorrhagic brain injury in a rodent model of bacterial collagenase-induced ICH (Figure 8). Specifically, intranasal delivery of Exo-124 attenuates neurological deficits, brain edema, BBB leakage and cell death after ICH. Importantly, Exo-124 reduces the counts of brain-infiltrating immune cells and decreases inflammatory cytokine levels in the brain after ICH onset. In addition, Gr-1^+^ myeloid cell depletion diminishes the protective effect of Exo-124 treatment. Our findings suggest that Exo-124 may be a potential therapeutic candidate for improving outcomes following ICH-induced brain injury.

Increasing evidence has demonstrated that miRNA-124 is a critical modulator of inflammation [41]. Previous studies by our group and others have confirmed that upregulation of miRNA-124 is involved in the transformation of microglia from a resting state or the proinflammatory phenotype to the anti-inflammatory phenotype [21,42,43]. Exo-124 protects against spinal cord injury by inhibiting neurotoxic glial activation [16]. Additionally, evidence suggests that the gene expression of miRNA-124 in the perihematomal tissue is significantly decreased after ICH [42]. Consistent with previous findings, our results suggest that Exo-124 may exert a protective effect against neuroinflammation after ICH.

In addition, brain-derived exosomes play an essential role in regulating the pathogenesis of CNS diseases. Previous studies have demonstrated that brain-derived exosomes are similarly involved in the regeneration and remodeling of the nervous system [44]. Microglia-derived exosomes, which carry enzymes and membrane receptors, were previously found in dendritic cells and B cells by mass spectrometry analysis; this finding is consistent with the roles of dendritic cells and B cells in the immune system [45]. Studies have indicated that activated microglia promote the release of exosomes carrying misfolded proteins, such as α-synuclein, tau and beta-amyloid, in neurodegenerative diseases [7,46,47]. Previous research on microglia-derived exosomes has mainly focused on the role of these vesicles in the development of chronic diseases, especially neurodegenerative diseases. This study elucidated the therapeutic effect of miRNA-edited microglia-derived exosomes on acute ICH from a new perspective. The therapeutic application of miRNAs is limited by the short half-life and poor systemic stability of miRNAs [48]. Reportedly, due to their double-layered membrane structure, exosomes contribute to preventing cargo clearance through binding to complements. Intranasal delivery of exosomes has attracted increasing attention because it is a noninvasive approach with low risks that can be used to target the brain. Intranasal drug delivery provides direct transport into the brain via the olfactory or trigeminal pathway. In addition, intranasal drug delivery provides distinct advantages, including high bioavailability, low dose, quick onset of action and high patient compliance. Of note, the application of intranasal administration is restricted to the inherent features of the nasal cavity, resulting in drug retention and irreversible entrapment of the drug [36,49,50]. In the present study, we designed miRNA-edited microglia-derived exosomes and administered them intranasally to achieve therapeutic concentrations in the brain. 

To our knowledge, brain inflammation and BBB dysfunction contribute to the exacerbation of PHE after ICH onset [51,52]. A previous study indicated that microglia-derived exosomes protect vascular endothelial cells from oxidative stress and improve outcomes following spinal cord injury [53]. Thus, we postulate that the neuroprotective effect of Exo-124 after ICH is related to immune cell infiltration into the brain. In support of this view, our findings demonstrate that the numbers of neutrophils and monocytes decrease rapidly following Exo-124 treatment after ICH. Intriguingly, Gr-1^+^ myeloid cell depletion significantly diminishes the neuroprotective effect of Exo-124, suggesting that the protective effect of Exo-124 involves Gr-1^+^ myeloid cells.

Although the present study demonstrates that exosomes are promising noninvasive agents for ICH treatment, there are some limitations to this study. Firstly, primary microglia would be the best model of the study. However, production of exosomes isolated from primary microglia in vitro is limited, which is difficult to achieve in our experiment. In addition, to better determine the transfection efficiency of miRNA-124 in microglia and its exosomes, the expression of miRNA-124 in the non-exosome fraction needs to be measured. Secondly, the exact mechanisms by which Exo-124 inhibits brain inflammation after ICH remain to be further clarified. To answer these questions, we are currently exploring the downstream target genes and related signaling pathways of miRNA-124 using bioinformatics analysis. However, the side effects, safety, and pharmacokinetic parameters of exosome therapy are not fully clear. Specific labeling methods, including GFP reporter, glucose-coated gold nanoparticle and fluorescent dyes, contribute to the distribution and monitoring of exosomes. Moreover, the role of miRNA-124 on microglia needs to be further clarified, such as the alteration of morphology, phenotype and inflammatory mediators. The present study suggests that Exo-124 has a neuroprotective effect during the acute phase of ICH. Further studies are required to determine the extent of the effect of Exo-124 on brain recovery after ICH. In addition, female mice were not involved in this research. The therapeutic effects of Exo-124 in different sexes have not been fully verified. Hence, female mice should be studied to reduce experimental bias caused by gender factors in further studies.

Moreover, emerging evidence indicates that modification of exosomes with chemical or biological molecules may enhance exosome production and the drug concentration in target organs [54,55,56]. Similarly, engineering of exosomes may prolong their half-life in the circulation and improve biological activity. Finally, the exact transport and clearance routes of exosomes upon intranasal delivery require further investigation [57]. From the translational medicine perspective, additional research is required to explore the side effects of intranasal delivery of exosomes.

## 5. Conclusions

We demonstrate that intranasal delivery of Exo-124 significantly attenuates hemorrhagic brain injury and neuroinflammation by decreasing the infiltration of immune cells into the brain following the onset of ICH. In conclusion, our findings show that gene-edited exosomes are an appealing nanotherapeutic agent for inhibiting the neuroinflammatory response.

## Figures and Tables

**Figure 1 brainsci-13-00639-f001:**
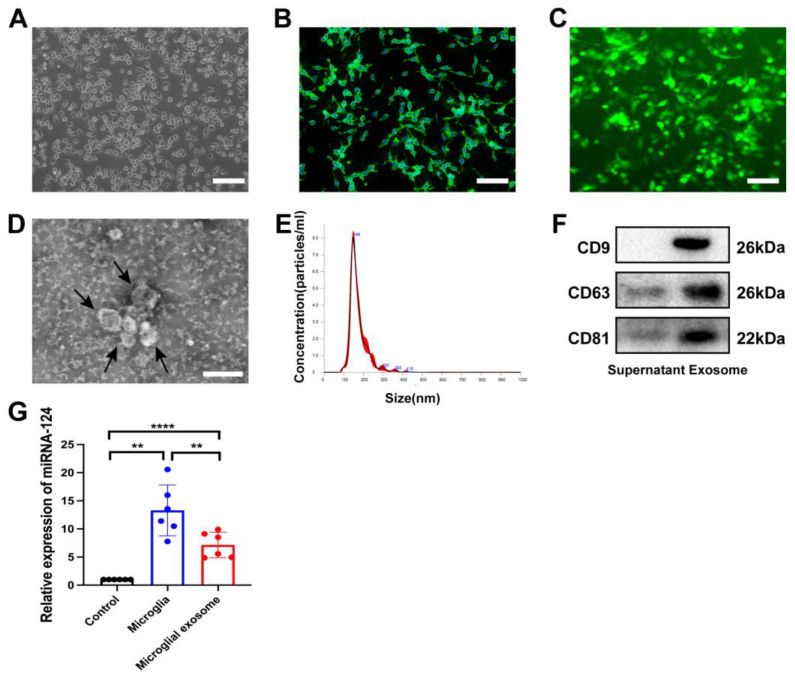
Characteristics of miRNA-124-enriched microglia-derived exosomes. (**A**) Representative light microscope image of cultured BV2 microglia. Scale bar: 200 µm. (**B**) Immunofluorescence staining of Iba-1 (green) and DAPI (blue) to identify microglia. Scale bar: 100 µm. (**C**) Representative image of miRNA-124-GFP (green)-transfected BV2 microglia. Scale bar: 100 µm. (**D**) Representative TEM image of microglia-derived exosomes. The black arrows indicate microglia-derived exosomes. Scale bar: 200 nm. (**E**) The size distribution of the microglia-derived exosomes was determined by a nanoparticle-tracking analyzer. (**F**) Western blot analysis of the levels of characteristic exosome biomarkers, including CD9, CD63 and CD81. (**G**) The bar graphs show the changes in miRNA-124 levels in microglia and microglia-derived exosomes after lentivirus transfection. n = 6/group, mean ± SD, ** *p* < 0.01,.**** *p* < 0.0001.

**Figure 2 brainsci-13-00639-f002:**
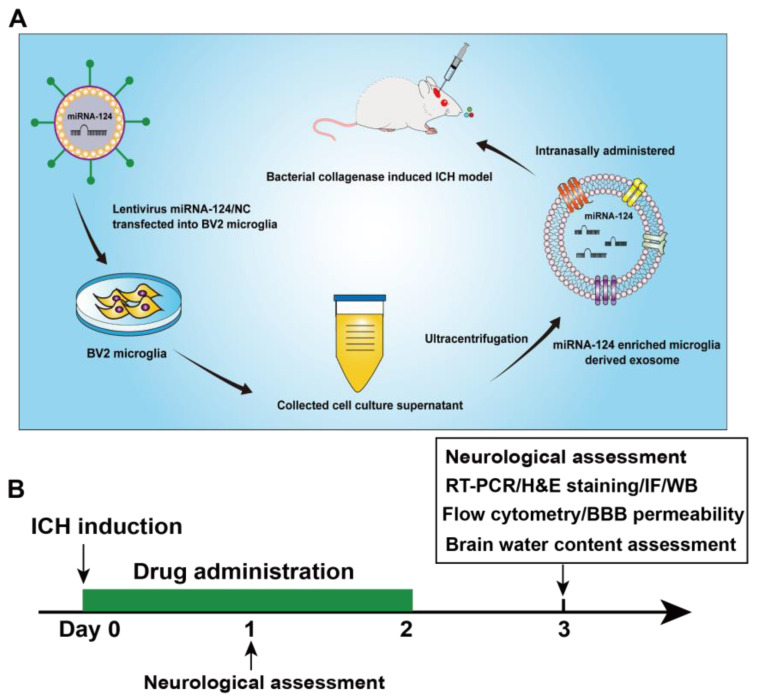
Schematic of the experimental design. (**A**,**B**) The schematic shows the experimental process, including exosome preparation, animal model establishment, and exosome administration.

**Figure 3 brainsci-13-00639-f003:**
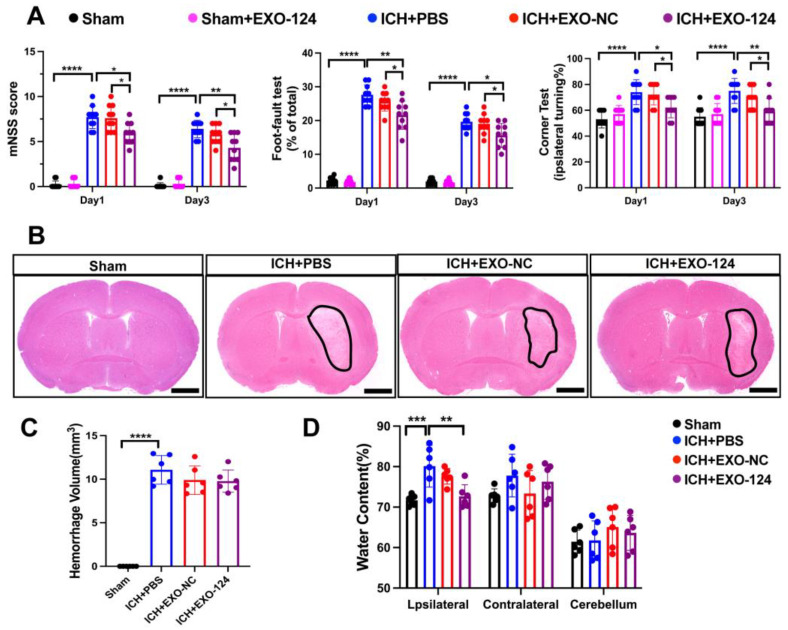
miRNA-124-enriched microglia-derived exosomes attenuate neurological deficits, brain edema and hematoma volume after ICH. (**A**) Summary of mNSS score and performance in the foot-fault test and corner turning test on days 1 and 3 after ICH induction. n = 10/group, mean ± SD, * *p* < 0.05, ** *p* < 0.01, **** *p* < 0.0001. (**B**) H&E staining was used to assess the hematoma volume in the regions outlined in black. Scale bar: 1 mm. (**C**,**D**) Quantification of the hematoma volume and brain water content at day 3 after ICH. n = 6/group, mean ± SD, ** *p* < 0.01, *** *p* < 0.001, *****p* < 0.0001.

**Figure 4 brainsci-13-00639-f004:**
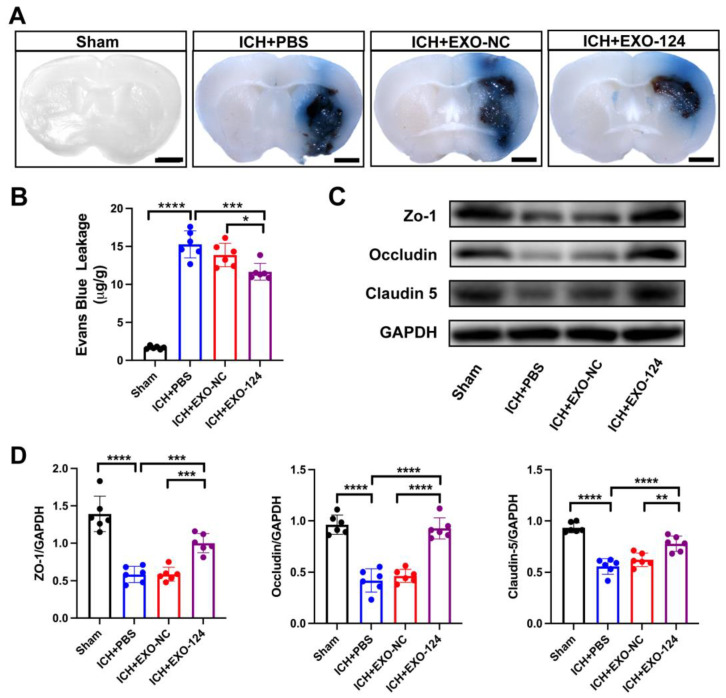
miRNA-124-enriched microglia-derived exosomes attenuate BBB damage. (**A**,**B**) Representative histological images and bar graphs showing that Evans blue dye leakage was decreased in the Exo-124 treated group compared with the PBS-treated group. Scale bar: 1 mm. n = 6/group, mean ± SD, ** *p* < 0.01. (**C**,**D**) Western blot showing that the levels of tight junction proteins, including claudin-5, occludin and ZO-1, were higher in the Exo-124-treated group. n = 6/group, mean ± SD, * *p* < 0.05, ** *p* < 0.01, *** *p* < 0.001, **** *p* < 0.0001.

**Figure 5 brainsci-13-00639-f005:**
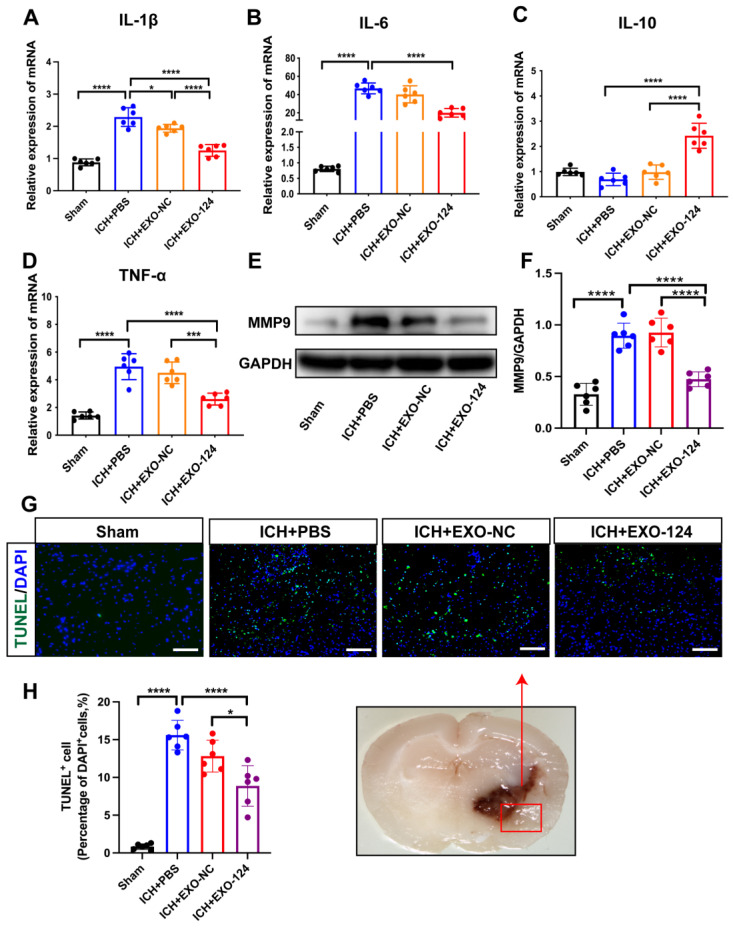
miRNA-124-enriched microglia-derived exosomes reduce cell death and inhibit brain inflammation after ICH. (**A**–**D**) Real-time PCR analysis of the mRNA expression of inflammatory cytokines (TNF-α, IL-1β, IL-6 and IL-10). n = 6/group, mean ± SD, * *p* < 0.05, *** *p* < 0.001, **** *p* < 0.0001. (**E**,**F**) Immunoblotting and quantitative data of MMP9 after Exo-124 treatment. n = 6/group, mean ± SD, **** *p* < 0.0001. (**G**,**H**) Representative images showing TUNEL^+^ cells in the indicated groups. The red rectangle indicates the measured region. The bar graph shows the quantitative data. Scale bars: 100 µm. n = 6/group, mean ± SD, * *p* < 0.05, **** *p* < 0.0001.

**Figure 6 brainsci-13-00639-f006:**
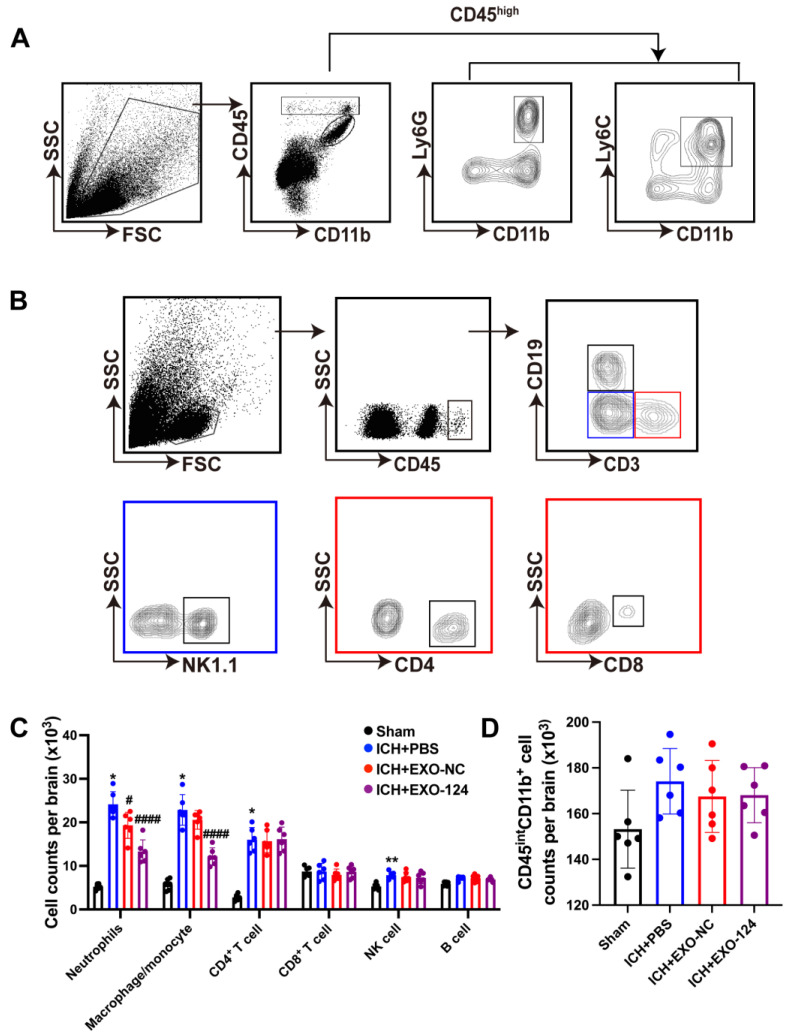
miRNA-124-enriched microglia-derived exosomes reduce immune cell subset infiltration into the brain after ICH. (**A**,**B**) Representative flow cytometry plots showing the gating strategies for brain-infiltrating immune cells, including CD4^+^ T cells (CD45^high^CD3^+^CD4^+^), CD8^+^ T cells (CD45^high^CD3^+^CD8^+^), B cells (CD45^high^CD3^−^CD19^+^), NK cells (CD45^high^CD3^−^NK1.1^+^), neutrophils (CD45^high^CD11b^+^Ly6G^+^), monocytes/macrophages (CD45^high^CD11b^+^Ly6C^+^) and microglia (CD45^int^CD11b^+^). (**C**,**D**) The bar graph shows that the infiltration of neutrophils and monocyte/macrophage cells was decreased in the Exo-124-treated group compared with the PBS-treated group. n = 6/group, mean ± SD. * *p* < 0.05, ** *p* < 0.01 vs. the sham group, ^#^ *p* < 0.05, ^####^ *p* < 0.0001 vs. the ICH + PBS group.

**Figure 7 brainsci-13-00639-f007:**
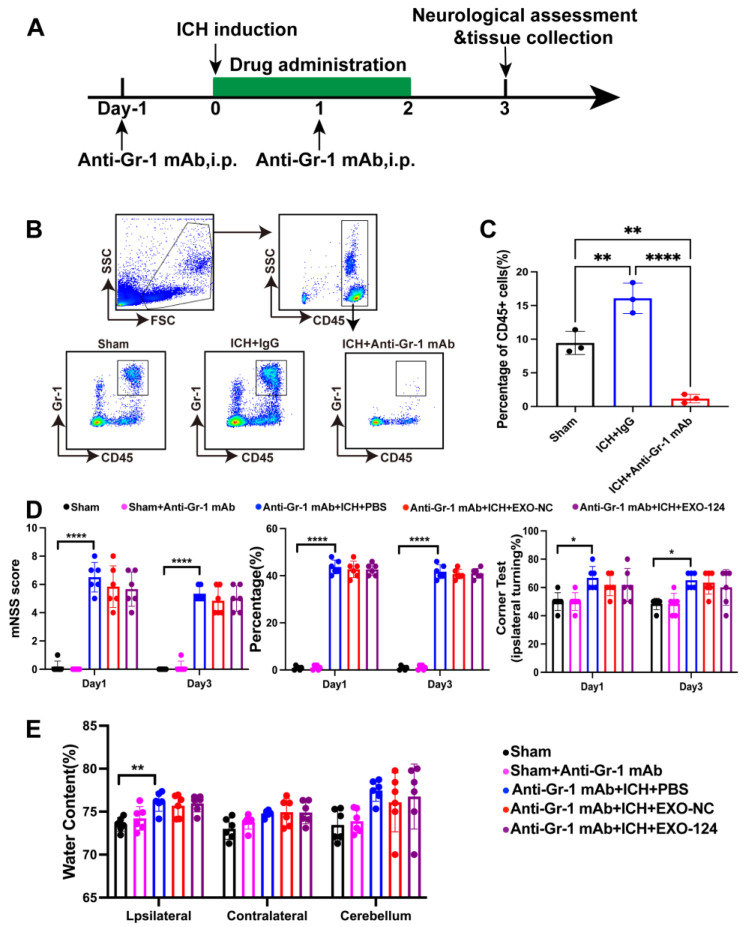
Gr-1^+^ myeloid cells are involved in the protective effect of miRNA-124-enriched microglia-derived exosomes. (**A**) Flow chart showing the drug administration protocol and experimental design. (**B**,**C**) Representative flow cytometry plots showing the gating strategy for circulating Gr-1^+^ cells in mice. n = 3/group, mean ± SD, ** *p* < 0.01, **** *p* < 0.0001. (**D**) Summary of mNSS score and performance in the foot-fault test and corner turning test on days 1 and 3 after ICH induction following administration of Gr-1 mAb or the indicated. n= 6/group, mean ± SD, * *p* < 0.05, **** *p* < 0.0001. (**E**) The bar graph shows the brain water content in ICH mice that received Gr-1 mAb or the indicated treatment. n = 6/group, mean ± SD,** *p* < 0.01.

**Figure 8 brainsci-13-00639-f008:**
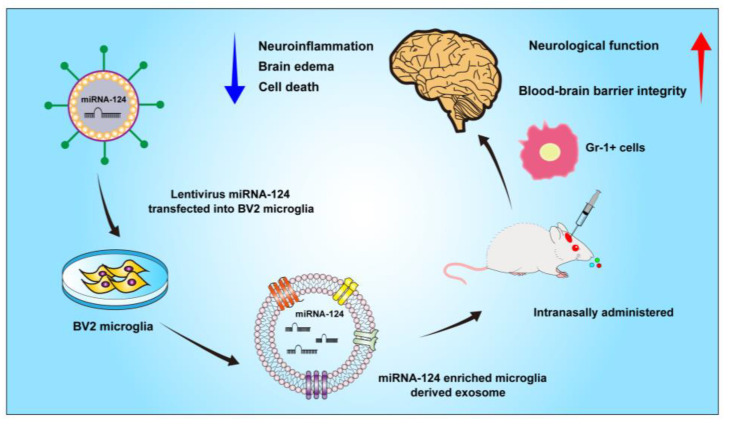
Summary of the effect of miRNA-124-enriched microglia-derived exosomes after ICH. Lentivirus vectors expressing miRNA-124 were transduced into BV2 microglia. The cell culture supernatant was collected and ultracentrifuged. ICH was induced in mice by bacterial collagenase, and the mice received miRNA-124-enriched microglia-derived exosomes via intranasal administration. miRNA-124-enriched microglia-derived exosomes reduced brain edema, cell death and neuroinflammation in ICH mice. miRNA-124-enriched microglia-derived exosomes improved BBB integrity and neurological function. The beneficial effects of miRNA-124-enriched microglia-derived exosomes were associated with amelioration of inflammatory milieu and a reduction in Gr-1^+^ myeloid cell infiltration into the brain.

**Table 1 brainsci-13-00639-t001:** Primer sequences of real-time PCR.

Primers	Forward Sequences (5′-3′)	Reverse Sequences (5′-3′)
IL-1β	TCGCAGCAGCACATCAACAAGAG	AGGTCCACGGAAAGCACACAGG
IL-6	ACGCTTCTGGGCCTGTTGTT	CCTGCTGCTGGTGATTCTCT
IL-10	TCCCTGGGTGAGAAGCTGAAGAC	CACCTGCTCCACTGCCTTGC
TNF-α	GCCTCTTCTCATTCCTGCTTGTGGG	GTGGTTTGAGTGTGAGGGTCTG
GAPDH	GCCAAGGCTGTGGGCAAGGT	TCTCCAGGCGGCACGCAGA
miRNA-124	TCTTTAAGGCACGCGGTG	TATGGTTTTGACGACTGTGTGAT
U6	CTCGCTTCGGCAGCACA	AACGCTTCACGAATTTGCGT

## Data Availability

The data presented in this study are available on request from the corresponding author.

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
