# Peer review of "Intranasal Delivery of Gene-Edited Microglial Exosomes Improves Neurological Outcomes after Intracerebral Hemorrhage by Regulating Neuroinflammation"

_brainsci, 2023, doi:10.3390/brainsci13040639_

Round 1

Reviewer 1 Report

In the study of Guo et al, the authors investigated the effect of intranasal injections of miRNA-124-enriched exosomes on cerebral damage in a rat model of ischemic hemorrhage. To do this, they isolated exosomes from BV2 microglial cells transfected with the cDNA of miRNA-124. The intranasal administration of miRNA-124 exosomes allowed observing functional recovery improvement and smaller histological lesions in the brain. The injury-induced BBB leakage was reduced. Along with this, the expression of inflammatory cytokines (e.g. TNFa) and the infiltration of some immune cells (neutrophils and monocytes) were decreased. Taken together, this study reports interesting protective effects of intranasally-administered miRNA-124 in exosomes in a model of cerebral ischemia. The approach used in this preclinical model has a strong translational potential. That said, I think that the authors should document the uptake and biodistribution of miRNA-124/exosomes using the intranasal route of administration. Indeed, the nose-to-brain transport of exosomes is not trivial. The authors used 100 mg of miRNA-124, a very high amount. Therefore, the question relates to the actual amount of miRNA-124 that can reach the brain and the proportion that goes to the blood flow/peripheral organs. In addition, as the BBB is compromised after ischemia, the advantage of using the intranasal pathway for exosomes delivery, vs the intravenous pathway, is not obvious. To monitor miRNA-124/exosomes, the authors could use the GFP reporter that is co-expressed with miRNA-124. The expression of genes that can potentially be influenced by miRNA-124 should be considered; in regard with the side effects, that miRNA-124 may have in the brain, and to clarify the mechanism of action mitigating inflammation.

Reviewer 2 Report

The authors of this manuscript, Guo et al., attempted to investigate the potential therapeutic effects of miRNA-124 enriched microglia exosomes on intracerebral hemorrhage in the present study. I would like to express my appreciation to the writers for the skillful construction of the manuscript, which has a flow that is engaging for the audience. However, I have some concerns about the manuscript.

Minor concerns:

  1. How was the intranasal dose for the mice determined? Are there any previous dose-response studies that have been done? If yes, please mention it on the manuscript.
  2. The authors did not show or mention how stable these exosomes are. Freshly isolated exosomes were used or, after storage, they were used. Freeze-thaw stability is required for the exosomes.
  3. Any specific reason why commercial BV-2 cell lines were used and not cells isolated from C57BL/6 pups?
  4. Is there any reason why adult male C57BL/6 mice were used and not females? Do we see any gender difference in the ICH mouse model (bacterial collagenase)? The extent of damage is the same?

Reviewer 3 Report

Overall, this study is interesting.  Considering the complexity of the neurobiology of ICH, the heterogeneous patient population, and the limitations of currently available treatments to alleviate neuroinflammation, there is a need for more studies to address these limitations. In this regard, the findings of the current study are significant and will open possibilities for developing new exosome mediated therapies.

However, there are a lot of errors in the manuscript and some areas are not clear. Addressing these shortcomings is very important before consideration for publication. Below are some of those concerns and suggestions.

The authors introduced the terminology Exo-124 suddenly in the end of the introduction. Please include that in the abstract as well. Also give some description for the EXO-124 in the introduction and that will be easy to understand.

Why only male mice used for this study? Complete study should have both sexes. Please explain.

The authors used the BV2 cell line to isolate the exosomes. Since it’s a cancer cell line the exosome composition might be have been vary with Primary microglia. Primary microglia would have been the best model for this study.

The role of mir-124 has been not described clearly. Also, what happens when the mir-124 overexpressed in the BV2 cells? Any changes in the morphology, phenotype? What was the levels of proinflammatory cytokines levels in the mir-124 overexpressed exosomes and as well as in the cell culture media?

Please explain how they measured the exosome in ug? Please describe the methods

Fig 1F there is no lane marking. Please complete it.

Fig 1G & H why the mir-124 expression difference huge between microglia and exosomes? Is all the mir-124 secreted outside of the cell via exosomes? Please show the expression of mir-124 in the non-exosome fraction.  

Overall, the Invivo study design should be improved. There is no sham control in the most of the experiments. Without sham control the real effect of Exo-124 will not be evaluated.

Fig 3A-D include the sham control.

In the method section, please explain detailly how the mNSS score was measured from different behavior analysis.

Fig 4A-D include shame control in all the experiments.

Fig 5 Sham control included only in Fig 5A,B,C and D. why not in other experiments. Please include.

Please provide all the raw images of the western blots.

In Fig 3-5 include the EXO-124 group alone. It will very interesting to see the Exo-124 effect in normal animals. Please provide the behavioral data at least.

Again Fig 7 Sham control missing. Please include.

What about the Anti Gr-1 mAb alone group? Please include the group and provide the behavioral outcomes after the treatment.

Is authors found any significant change Gr-1+ myloid cells number or markers expression during or after the ICH induction?

Round 2

Reviewer 2 Report

The authors justified the comments well and gave proper references. I feel the manuscript is suitable for publication.

Author Response

Thank you for your reply and support.

Reviewer 3 Report

Dear Authors,

Thanks for your explanation for the concerns raised. The most of the explanations are convincing but still there are few concerns need to be addressed. I have given below the previous concerns which is not well addressed.

1.      In 2.5 mentioning 2.11 × 118 109± 1.04 × 107particles/ml would be appropriate than 100ug of exosome. Because 100ug is a protein concentration not the whole exosome content. In that case how authors claim the protein concentration as an exosome quantity? Because exosome packed not only proteins, but they also contain RNA, DNA etc.

2.      Fig 1G & H why the mir-124 expression difference huge between microglia and exosomes? Is all the mir-124 secreted outside of the cell via exosomes? Please show the expression of mir-124 in the non-exosome fraction. Authors replied “The expression of miRNA-124 in the non-exosomes fraction is required using RT-PCR in the further experiment” do authors have any plan to do that experiment?

3.      For all western blot presented data, the authors must provide with exact number of the replicas and experiments used for quantifications and statistical analyses, images of all membranes (particularly when juts one band is show per condition) must be added as supplementary material. They also should mention if the presented images in a panel are all coming from the same membrane, if it is the case include membrane stripping protocol in the methods section, and if it is not the case, provided with all loading controls in the figure panels.
